# Ultrafast charge-transfer-induced spin transition in cobalt-tungstate molecular photomagnets

Kazuki Nakamura [1,2], Laurent Guérin [2,3], Gaël Privault [2,3], Koji Nakabayashi [1,2] ✉, Marius Hervé [2,3], Eric Collet [2,3,4] ✉ & Shin-ichi Ohkoshi [1,2] ✉

In materials exhibiting photoinduced phase transitions, and in which both charge transfer and spin transitions occur, there has long been a debate about which process drives the phase transition. Herein, we present experimental evidence supporting an optically charge-transfer-induced spin transition (CTIST) process, as demonstrated through femtosecond optical spectroscopy in two-dimensional cyanido-bridged cobalt-tungstate photomagnets. Optical and magnetic studies revealed that the photoexcitation of the ground low-temperature (LT) $Co^{III}_{LS}$-$W^{IV}$ state leads to a photoinduced phase transition towards the $Co^{II}_{HS}$-$W^{V}$ state, which is similar to the high temperature (HT) state. Ultrafast spectroscopy further indicates that this optical excitation of the intermetallic W-to-Co charge-transfer band produces a transient photoexcited (PE) $Co^{II}_{LS}$-$W^{V}$ state, which decays within 130 fs through a spin transition towards the $Co^{II}_{HS}$-$W^{V}$ state. Here we show that the CTIST dynamics corresponds to the $Co^{III}_{LS}$-$W^{IV}$ (LT) → $Co^{II}_{LS}$-$W^{V}$ (PE) → $Co^{II}_{HS}$-$W^{V}$ (HT) sequence. The present work sheds a new light on understanding optical dynamics underlying the photoinduced phase transitions.

The rational design of molecular materials[1–5] aims to develop and control their physical properties for specific applications, posing a substantial challenge in material science, chemistry, and physics. Phase transitions in molecular materials, which involve changes in various physical properties[6–8], can be regulated by external stimuli of chemical (solvent, pH)[9–11] and physical (temperature, current, pressure, and light) nature[12–15]. Photoinduced phase transitions (PIPTs) represent a promising avenue for switching physical properties by altering electronic and structural degrees of freedom under photoirradiation[16–18]. Ultrafast time-resolved optical techniques allow to gain substantial knowledge in the understanding of the photoinduced processes at play in PIPTs, enabling the study of electronic and structural dynamics[19–22] at the molecular scale and the complex and multiscale out-of-equilibrium transformations at the macroscopic scale[23–25]. This research has facilitated advancements in applications, such as photonic actuators, memory devices, and other photonic technologies[17,26].

Cyanido-bridged heterometallic assemblies are promising molecular materials because of their ability to exhibit electronic changes driven by coupled charge transfer (CT) and/or spin transition (ST), which can be triggered by temperature or light[27–30]. These transitions enable functional switching of magnetic, optical, and thermodynamical properties[31–34]. Among the heterometallic assemblies, cyanido-bridged cobalt-tungstate assemblies show phase transitions with a thermal hysteresis loop and photomagnetism originated from an optical transition between the low-spin $Co^{III}_{LS}$-$W^{IV}$ low-temperature (LT) state and the high-spin $Co^{II}_{HS}$-$W^{V}$ high-temperature (HT) state

[1]Department of Chemistry, School of Science, The University of Tokyo, 7-3-1 Hongo, Bunkyo-ku, Tokyo, Japan. [2]DYNACOM IRL2015 University of Tokyo - CNRS - Université de Rennes, Department of Chemistry, 7-3-1 Hongo, Bunkyo-ku, Tokyo, Japan. [3]Univ Rennes, CNRS, IPR (Institut de Physique) - UMR 6251, Rennes, France. [4]Institut Universitaire de France (IUF), Paris, France. ✉e-mail: knakabayashi@chem.s.u-tokyo.ac.jp; eric.collet@univ-rennes.fr; ohkoshi@chem.s.u-tokyo.ac.jp

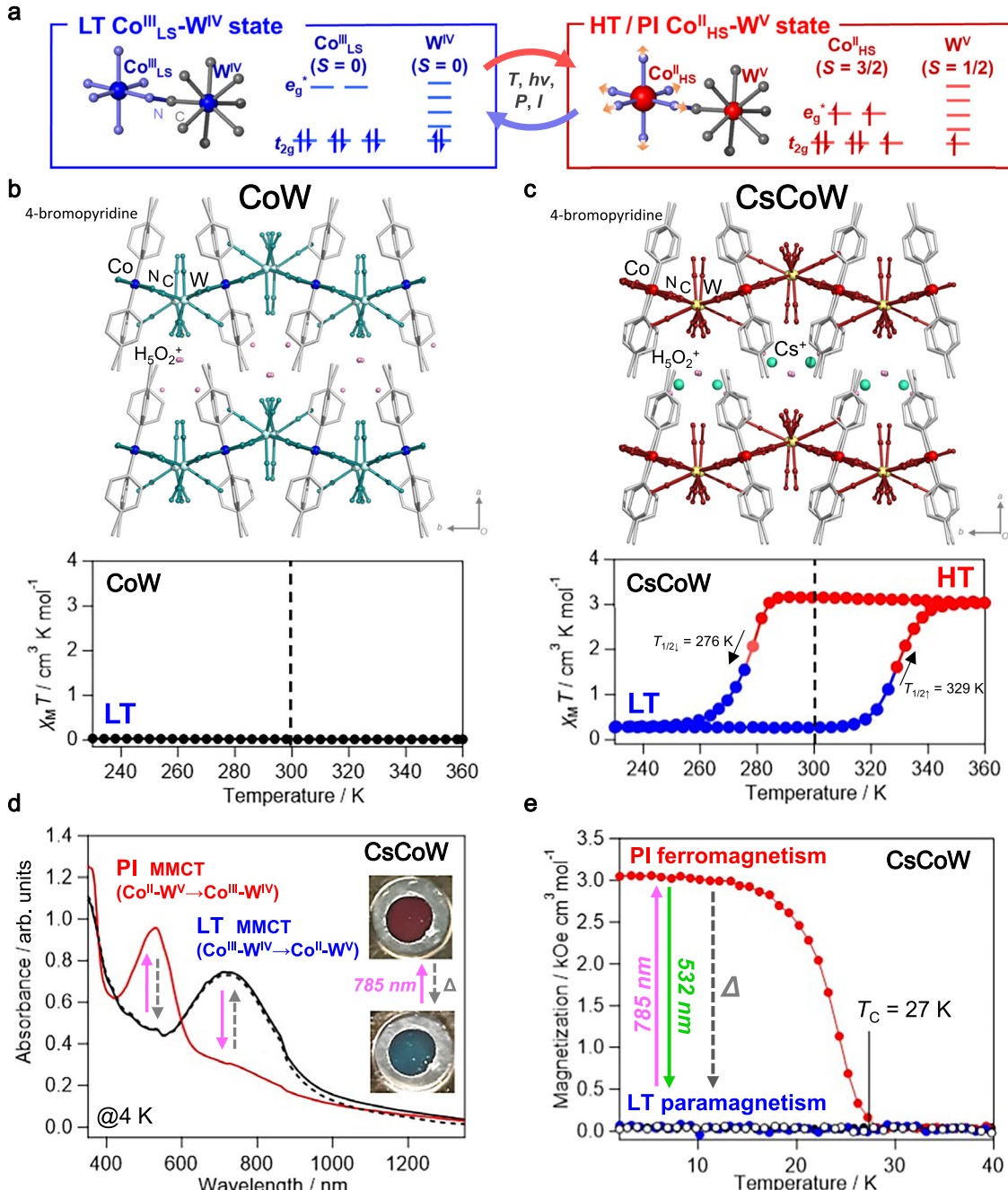

**Fig. 1 | Magnetic and optical properties of cyanido-bridged cobalt-tungstate assemblies. a** A schematic representation of the cyanido-bridged cobalt-tungstate moiety, illustrating low-spin $Co^{III}_{LS}(S = 0)$ $W^{IV}(S = 0)$ low temperature state (LT, blue) state and high-spin $Co^{II}_{HS}(S = 3/2)$-$W^V(S = 1/2)$ high temperature (HT) or photo-induced (PI) states (red). The spin transition on the Co site in the HT or PI states reduces the $t_{2g}$-$e_g$ splitting, accompanied by elongation of the Co–N bonds (orange arrows). **b, c** The crystal structures and $\chi_M T$ vs $T$ plots for CoW and CsCoW, respectively, reproduced with permission from references. [38] and [39] (RSC). CoW remains in the LT state[38], whereas CsCoW undergoes a thermal transition between HT (red) and LT (blue) states[39]. The crystal structures depict cyanido-bridged Co-W layers and interlayer ions (Cs⁺: green, $H_5O_2^+$: pink). **d** UV-vis-NIR spectra of CsCoW were measured at 4 K before irradiation (black line), after irradiation (red line; 785 nm, 160 mW cm⁻²), and after thermal treatment at 100 K (dotted black line). The inset shows images of the drastic color change before (bottom, LT state) and after irradiation (top, PI state, red). **e** Photomagnetism behavior of CsCoW. Field-cooled magnetization curves under 20 Oe show the paramagnetic LT state (black circles). Photoinduced magnetization is observed at 785 nm (240 mW cm⁻², 3 K; red), which thermally relaxes to the LT state above 100 K (open black circles). Photoexcitation at 532 nm (145 mW cm⁻², 3 K) switches the system to the para-magnetic LT state (blue).

(Fig. 1a)[35–37]. These electronic transformations are further associated with the drastic spectral change in the distinct ultraviolet-visible-near-infrared (UV-vis-NIR) absorption due to the metal-to-metal CT (MMCT) from $Co^{III}$-$W^{IV}$ to $Co^{II}$-$W^V$ in the LT or vice versa in the HT states. Their optical and magnetic properties and phase transitions can be tunable by varying the counter ions, solvents, and organic ligands. For

instance, the cyanido-bridged cobalt-tungstate assembly, $(H_5O_2^+)$ $[Co(4\text{-bromopyridine})_2\{W(CN)_8\}]$ (CoW) exhibits a stable LT state across a wide temperature range, extending beyond room temperature (RT) (Fig. 1b)[38]. In contrast, the partially-Cs-substituted cobalt-tungstate assembly, $Cs^+_{0.1}(H_5O_2^+)_{0.9}[Co(4\text{-bromopyridine})_{2.3}\{W(CN)_8\}]$ (CsCoW), exhibits a LT–HT thermal phase transition with LT–HT

bistability at RT along with an 8% volume expansion (Fig. 1c)[39]. The partial substitution indeed destabilizes the hydrogen-bond network between the layers, resulting in a more flexible lattice that stabilizes the higher-volume high-spin HT state at RT.

In PIPT materials involving both CT and/or ST[40–42], there has been a longstanding debate spanning approximately 30 years on which of the two processes is leading the phase transition: Charge-transfer-induced Spin transition (CTIST), or vice versa (Spin transition-induced Charge transfer: STICT)? Ultrafast pump-probe techniques were used to investigate the photoinduced process[43–45]. Recently, an optically STICT process was reported in a CoFe Prussian blue analogue[46]. In contrast, density functional theory (DFT) calculations for cyanido-bridged cobalt-tungstate assemblies indicate that optical excitation of the MMCT band in the LT state corresponds to electron transfer from W to Co[38]. In the present work, we examine the ultrafast photoinduced dynamics in CsCoW at RT by comparing with the ultrafast spectroscopic changes observed in CoW. Sub-picosecond (ps) and 10's ps dynamics studies demonstrate the photoinduced dynamics at molecular and lattice scales. Our results provide experimental evidence for CTIST in both CsCoW and CoW. This process occurs on sub-ps molecular dynamics, while a slower thermoelastic conversion is observed only in CsCoW, near the transition temperature and under high laser fluence, occurring on the 10's ps timescale.

## Results

### Photoinduced phase transition at low temperature

The PIPT of CsCoW at low temperatures was investigated using UV-vis-NIR absorption spectroscopy and a superconducting quantum interference device magnetometer. In the LT state, CsCoW exhibits an MMCT band from $Co^{III}$-$W^{IV}$ to $Co^{II}$-$W^{V}$, centered at 760 nm (Fig. 1d). The absorption band is comparable to the MMCT band of CoW, where the highest occupied crystalline orbital (HOCO) consists of $d_z^2$ orbitals of $W^{IV}$ and p orbitals of nitrogen (N) atoms. In contrast, the lowest unoccupied crystalline orbital (LUCO) comprises $d_z^2$ orbitals of $Co^{III}$ and sp orbitals of N atoms, as indicated by DFT calculations[38]. Therefore, a continuous-wave (cw) diode laser operating at 785 nm was used as the photoirradiation source. Fig. 1d illustrates the UV-vis-NIR spectra before and after photoirradiation at 4 K. Under photoirradiation, the absorption band around 760 nm, which is characteristic of the ground LT state, disappears, and the sample color changes from blue to red (Fig. 1d inset) as a new band emerges at 530 nm, which is a distinctive feature of the photoinduced (PI) state and corresponds to the MMCT band from $Co^{II}$-$W^{V}$ to $Co^{III}$-$W^{IV}$. This color change resembles the thermochromism observed during the LT to HT phase transition[38]. The MMCT band in the PI state is slightly red-shifted compared to the band at 495 nm in the HT state, which is attributed to the smaller energy gap due to changes in the ligand field of the Co site, mainly decreasing the π-back donation from the cyanide ligand, owing to the distortion of the cyanido-bridged network by the volume contraction at low temperatures (Supplementary Fig. 4a). Upon warming, the 530 nm peak of the PI state disappears while the 760 nm peak intensifies around 70 K, signifying thermal relaxation from the PI state to the LT state (Supplementary Fig. 4b). Additionally, reverse-PIPT was studied to confirm the photoreversibility from the PI state to the LT state using a cw diode laser at 532 nm, which aligns with the MMCT band of the PI state at 4 K. During irradiation of the PI state, the 530 nm peak decreases, and the 760 nm peak reappears (Supplementary Fig. 4c), directly evidencing the reverse-PIPT from the PI state to the LT state. These findings demonstrate that CsCoW undergoes a reversible-PIPT at LTs, exhibiting characteristics similar to those of CoW[38].

We conducted photomagnetic measurements at low temperatures, under the same irradiation conditions as the UV-vis-NIR measurements. Fig. 1e shows the field cooling magnetization (FCM) curves under 20 Oe. The LT state before photoirradiation is paramagnetic. Under 785 nm irradiation in the LT state at 3 K, the magnetization

increases due to the ferromagnetic nature of the PI state, characterized by a Curie temperature ($T_C$) of 27 K, the same as that reported for CoW. Both $J$ values of the superexchange interaction is 8.4 cm$^{-1}$, calculated by the molecular-field theory[36]. Since the $T_C$ strongly depends on the superexchange interactions between the metal spins through the cyanido-bridged network and their dimensionality, such as one-dimensional chain, two-dimensional layer, and three-dimensional network, the same $T_C$ indicates the similar superexchange interaction between Co and W in CsCoW and CoW, and remaining similar cyanido-bridged layer structures despite the partial cation substitution. The PI state returns to the initial paramagnetic LT state upon thermal treatment at 100 K, which is consistent with the UV-vis-NIR spectra. The magnetization versus magnetic field ($M$–$H$) plots at 2 K in the PI state show a coercive field of 7500 Oe, substantially larger than the 2000 Oe observed in CoW (Supplementary Fig. 5c). The saturated magnetization under 50 kOe was 3.1 $\mu_B$, aligning with the calculated value of 3.2 $\mu_B$, considering the ferromagnetic interaction between $Co^{II}$ ($g_{Co}$ = 13/3, $S_{Co}$ = 1/2 (Kramers doublet)) and $W^V$ ($g_W$ = 2.0, $S_W$ = 1/2) in the PI state. Remarkably, the PI state persisted for at least 1 day after the light was turned off at 3 K. The reverse-PIPT of the PI state was investigated using a 532 nm cw diode laser. As shown in Fig. 1e, the magnetization of the PI state at 3 K decreased to the initial value. Both photothermal and photoreversible PIPTs were cycled at least three times without any signs of degradation (Supplementary Fig. 5e, f). The FCM and $M$–$H$ curves also returned to the magnetization of the paramagnetic LT state. These measurements confirmed the photoreversible switching between the LT paramagnetic $Co^{III}_{LS}$-$W^{IV}$ and PI ferromagnetic $Co^{II}_{HS}$-$W^V$ states, which is similar to the HT state. Above 100 K, the lifetime of the PI state was too short to be observed using conventional techniques.

### General consideration of time-resolved optical spectroscopy

Femtosecond time-resolved optical spectroscopy was conducted to investigate the ultrafast dynamics underlying the PIPT through CT and ST processes in cyanido-bridged cobalt-tungstate assemblies. The visible region (500–740 nm)[47] was probed, where the drastic optical density change in CsCoW is characteristic of the electronic state switching. Measurements were performed above 100 K, where the short-lived PI state enables pump-probe measurements at a 1 kHz repetition rate. The sample was photoexcited with an approximate 65 fs laser pulse at 850 nm (8 mJ cm$^{-2}$). Two-time ranges of optical density changes ($\Delta OD(t)$) were analyzed to investigate distinct dynamics: ultrafast photoinduced dynamics (from −0.5 to 2.0 ps) and consecutive thermoelastic dynamics (from −50 to 800 ps). The dynamical responses in CsCoW and CoW were compared at RT, where the LT state of CoW and the LT state branch of the thermal hysteresis loop in CsCoW were photoexcited. Additionally, the temperature-dependent photoresponse of CsCoW was examined using a cryogenic system, with measurements conducted at 230 and 100 K, i.e., temperatures near and far from the thermal hysteresis loop, respectively.

### Time-resolved optical spectroscopy in sub-ps timescale

Figure 2 (top) shows the $\Delta OD(t)$ on the ps timescale, measured at RT following femtosecond laser excitation for CoW and CsCoW. In both cases, a decrease in OD was observed at 700 nm, corresponding to the strong bleaching of the LT $Co^{III}_{LS}$-$W^{IV}$ state. Additionally, a short transient increase in OD was evident around 600 nm for both compounds. The time trace $\Delta OD(t)$ at 600 nm remains positive in CsCoW, which can be attributed to the formation of the PI $Co^{II}_{HS}$-$W^V$ state. The $\Delta OD(t)$ data measured at 230 and 100 K for CsCoW resemble the data at RT for CsCoW and CoW, respectively, showing a transient OD increase near 600 nm and an OD decrease near 700 nm, accompanied by coherent oscillations (Supplementary Fig. 6). For both compounds, similar coherent oscillations are evident in the time traces, at the same

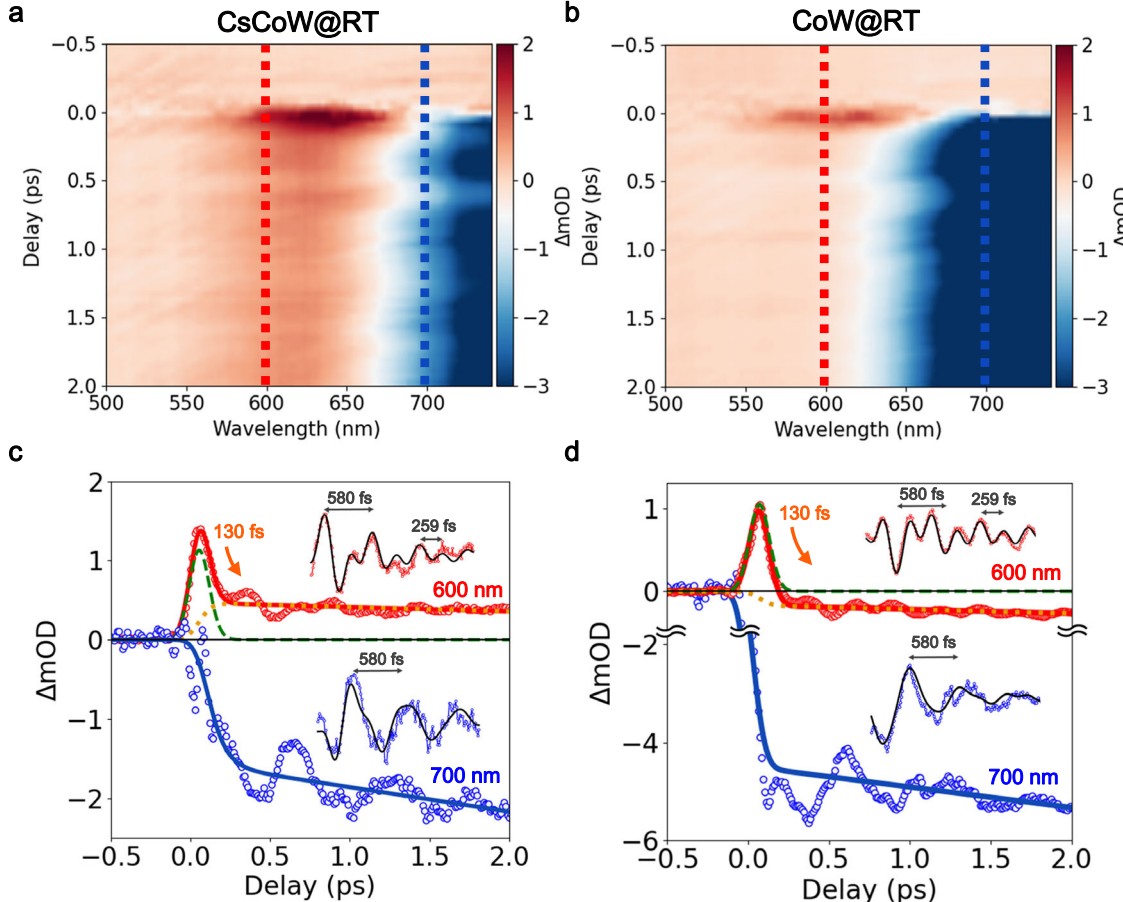

**Fig. 2 | Time-resolved optical spectroscopy on sub-ps timescale. a, b** Time delay ΔmOD (ΔOD ×10⁻³) maps measured at room temperature for CsCoW and CoW. Each map consists of three parameters: wavelength on the horizontal axis, delay time on the vertical axis, and ΔmOD as a color bar. The negative and positive delays indicate before and after the photoexcitation. The red and blue dashed lines represent the time traces at 600 nm and 700 nm, respectively. **c, d** Time traces of wavelength. ΔmOD at 600 nm (± 15 nm, red open circles) and 700 nm (± 15 nm, blue open circles) for CsCoW and CoW, respectively. The fit at 700 nm (blue) represents the exponential dynamics toward the photoinduced (PI) state, resembling the high temperature (HT) state. The fit at 600 nm (red) includes contributions from the PI state (orange) and an intermediate photoexcited (PE) state (green). The inset black line shows the oscillation fitting function. (Supplementary Note 6).

wavelength. In order to extract the characteristic timescales of the photoinduced dynamics, the OD changes were analyzed, resulting from the depopulation of the LT $Co^{III}_{LS}$-$W^{IV}$ state toward an initially electronic photoexcited (PE) $Co^{II}_{LS}$-$W^{V}$ state, identified by the intermediate transient peak around $t = 0$. Population fitting of $\Delta OD(t)$ at 600 and 700 nm was performed using an exponential decay model for the transient PE state, which subsequently populates the PI $Co^{II}_{HS}$-$W^{V}$ state (Fig. 2c, d, Supplementary Note 6). The fitting results indicate that the transient PE state decays within approximately 130 fs into the PI state, which is similar to the HT state. This decaying timescale is similar for CoW and CsCoW and comparable to that found in ST materials. In addition, it does not depend on the temperature, which is expected, given that it is driven by an electron-phonon coupling process rather than a thermally activated one[48]. This timescale is typical for photoinduced dynamics by ST, as reported in different spin-crossover materials exhibiting light-induced excited spin state trapping (LIESST effect) induced by metal-to-ligand CT photoexcitation[49–52]. Supplementary Fig. 7 shows the oscillating component of the $\Delta OD(t)$ signal after 300 fs, obtained as a residual difference between the experimental data and the fitted exponential functions (Supplementary Note 6). A fast Fourier transform (FFT) of this signal revealed the frequencies of these oscillating components. The wavelength *vs* wavenumber *vs* FFT intensity maps are similar for CoW and CsCoW (Supplementary Figs. 8,9). Two

common vibrational modes were identified for CsCoW, regardless of temperature: a fast oscillation with an approximate value of 259 fs period (approximately 130 cm⁻¹) and a slower oscillation with an approximate value of 580 fs period (approximately 58 cm⁻¹) (Supplementary Table 2). The spectral weight of the slower oscillation is more prominent at longer wavelengths. The oscillating signals at 600 and 700 nm, fitted using only these two nodes, closely match the experimental data (Fig. 2c, d insets, Supplementary Figs. 10, 11). At RT, $\Delta OD(t)$ remains positive near 600 nm after the transient peak, directly indicating the formation of the PI $Co^{II}_{HS}$-$W^{V}$ state in CsCoW. However, this OD increase is not clearly observed for CoW at RT or CsCoW at 100 K, which is likely due to the strong bleaching of the ground LT state absorption (Supplementary Fig. 14). A theoretical study discussed how the energy deposition on the molecule, through electronic excitation, drives consecutive electronic and structural dynamics on the sub-100 fs timescale and inter-system crossing, and the important role of the electron-phonon coupling[48]. The energy is released by electronic reorganization and the coherent and incoherent activation of molecular phonons through electron-phonon or phonon-phonon coupling. We could not investigate these faster dynamics in more detail because of our limited time resolution. In addition, optical spectroscopy does not allow the disentanglement of electronic and structural dynamics, as reported by ultrafast X-ray techniques in other materials[44–46].

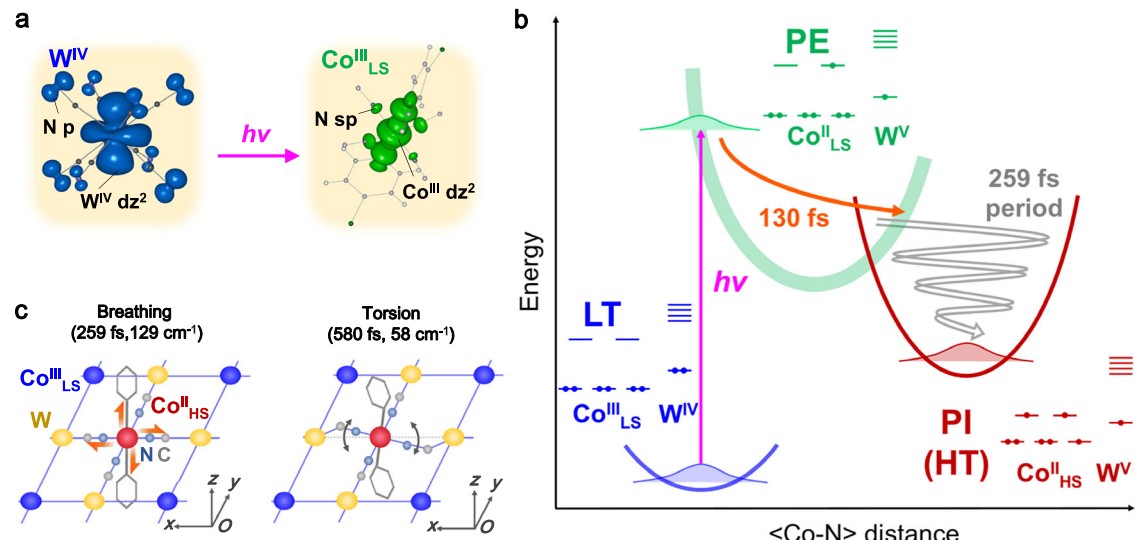

**Fig. 3 | Schematic representation of the potential energy curves and phonon modes. a** Charge density maps of the highest occupied crystalline orbital (HOCO, left) and lowest unoccupied crystalline orbital (LUCO, right) in the low-temperature (LT) state, reproduced from ref. 38 with permission from RSC. The calculated HOCO and LUCO filled the charge density on W and Co orbitals, respectively, indicating that the photoexcitation ($h\nu$) of the ground LT $Co^{III}_{LS}$-$W^{IV}$ state leads to the photoexcited (PE) $Co^{II}_{LS}$-$W^{V}$ state by promoting one electron transfer from W to Co orbitals. **b** The PE state decays within 130 fs towards the photoinduced (PI) $Co^{II}_{HS}$-$W^{V}$ state through the spin transition with the activation of the $CoN_6$ breathing mode. **c** Schematic representation of the phonon modes. The breathing mode (left) involves Co−N bond elongation (red arrows), while the torsion modes (right) represent distortions of Co−NC−W bridges (black arrows). During structural relaxation, Co−N bonds of the excited $Co^{II}_{HS}$ site elongate (approximately 130 cm$^{-1}$ phonon mode), causing lattice distortions and activation of the lattice torsion mode (approximately 58 cm$^{-1}$). The modes are shown in Supplementary Movie 1.

## Consideration of phonon modes

To understand the origin of the oscillations, we performed DFT calculations of the coordination environments of the Co site ($[Co(4\text{-}bromopyridine)_2(NC)_4]^{n-}$), which were evaluated for various electronic states: trivalent low-spin ($Co^{III}_{LS}$), divalent low-spin ($Co^{II}_{LS}$), and divalent high-spin ($Co^{II}_{HS}$) (Supplementary Note 7)[53]. Although these calculations were simplified by focusing solely on the Co environment, the phonon mode frequencies closely aligned with experimental values obtained from IR and THz spectra (Supplementary Fig. 12). The frequencies of similar phonon modes increased in the order $Co^{III}_{LS} < Co^{II}_{LS} < Co^{II}_{HS}$ (Supplementary Table 3). Phonon modes arising from cyanide ligands were observed below 500 and around 2100 cm$^{-1}$, and unique breathing modes were identified around 130 cm$^{-1}$ in $Co^{II}_{LS}$ and $Co^{II}_{HS}$, based on the structural optimization of $Co^{II}_{LS}$. These modes involve the stretching of organic and cyanide ligands. This type of mode is known to be strongly coupled with STs by changing drastic ligand fields between $t_{2g}$ and $e_g$ orbitals, as the modes act as the reaction coordinate. Consequently, these modes strongly affect metal-centered optical transitions. This type of coherent activation of breathing modes has been reported in other spin-crossover materials during the LIESST process[49–52]. DFT calculations also revealed a phonon mode at approximately 58 cm$^{-1}$, corresponding to a global torsion of the cyanido-bridged layer (Supplementary Movie 1). Such torsion modes have been previously reported in other Prussian blue analogues (Co−Fe and Mn−Fe systems)[54–56]. The ST leads to a local expansion of the cyanido-bridged network on a timescale too short for lattice expansion, activating torsion modes to accommodate these local distortions of the cyanide (Fig. 3c).

## Charge-transfer-induced spin transition in cyanido-bridged cobalt-tungstate assemblies

Based on femtosecond optical spectroscopy and DFT calculations, we propose that the photoinduced MMCT process involves a ground LT $Co^{III}_{LS}$-$W^{IV}$ and a PE $Co^{II}_{LS}$-$W^{V}$ states, because the STICT mechanism should be photoexcited the d-d transition of each metals, however, there are no such absorption bands in the

vis-NIR region, confirmed by the DFT calculations (Fig. 3a, Supplementary Fig. 13)[38]. Fig. 3b shows a schematic of the CTIST dynamics for both CsCoW and CoW, which share similar layered structures. The ultrafast photoinduced dynamics in cyanido-bridged cobalt-tungstate assemblies occur in two steps. First, optical excitation of the ground LT $Co^{III}_{LS}$-$W^{IV}$ state produces a PE $Co^{II}_{LS}$-$W^{V}$ state, characterized by a transient OD peak around 600 nm. This PE state is presumed to be stabilized by Jahn−Teller distortion and partial elongation of the average Co−N bonds, which can drive the ST. In the second step, by causing further elongation of the Co−N bond, the PE $Co^{II}_{LS}$-$W^{V}$ state decays within 130 fs into the lower-lying PI $Co^{II}_{HS}$-$W^{V}$ state. As discussed for the LIESST effect, the PE state functions as a mediator, with structural dynamics and electronic reorganization being entangled[57]. The timescale of this ST (approximately 130 fs) corresponds to half the period of the breathing mode associated with stretching of the C-N bonds and organic and cyanide ligands (approximately 259 fs) (Fig. 3b). This timescale is consistent with those reported for other Prussian blue analogues and ST materials[49–52,54–56]. On the 130 fs timescale of Co−N bond elongation trapping of the PE state, the crystalline lattice has no time to expand sufficiently, necessitating distortions to accommodate local lattice expansion. The ultrafast Co−N bond elongation launches consequently torsion modes of the W−Co−W cyanido-bridge, calculated at approximately 65 cm$^{-1}$ and consistent with the coherent oscillations observed at 58 cm$^{-1}$ (Fig. 3c). These sub-ps structural dynamics localized at the molecular scale are associated with the structural trapping of the PE state.

## Time-resolved optical spectroscopy on 10's ps timescale

We performed optical measurements on a longer ps timescale for the two compounds to monitor slower consecutive dynamics (Fig. 4). In all cases, the $\Delta OD(t)$ map around 700 nm reveals a rapid decay of the MMCT band, indicating ground-state bleaching immediately following photoexcitation. For CsCoW, an OD increase of around 600 nm was observed within 100 ps, characteristic of PI $Co^{II}_{HS}$-$W^{V}$ state

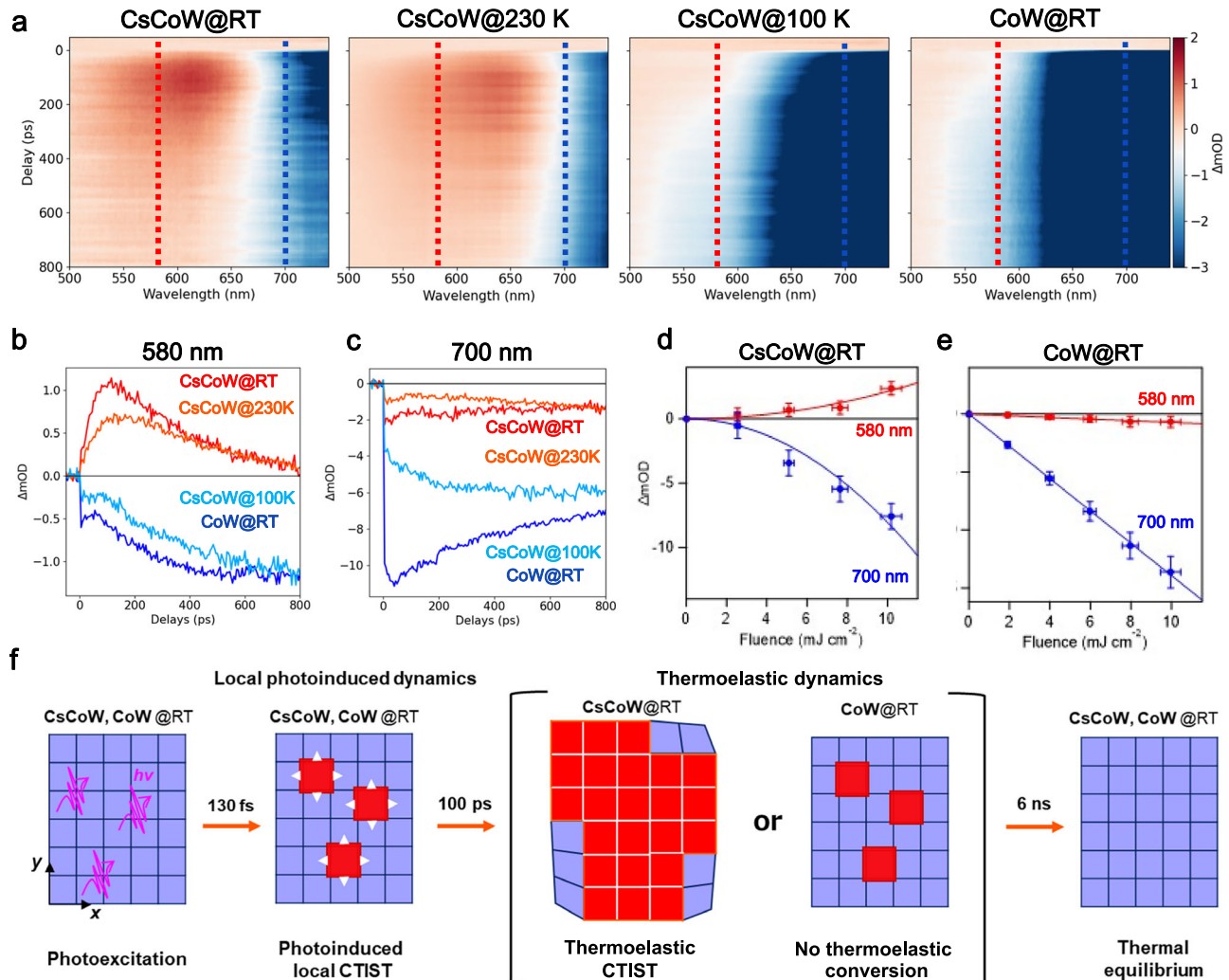

**Fig. 4 | Time-resolved optical spectroscopy on 10'ps timescale. a** Time delay ΔmOD (ΔOD ×10⁻³) maps for CsCoW at room temperature, 230, and 100 K and CoW at room temperature (8 mJ cm⁻²). Each map consists of three parameters: wavelength on the horizontal axis, delay time on the vertical axis, and ΔmOD as a color bar. The negative and positive delays indicate before and after the photoexcitation. The red and blue dashed lines represent the time traces at 580 nm and 700 nm, respectively. **b, c** Corresponding time traces at 580 nm (± 20 nm) and 700 nm (± 20 nm), respectively. **d, e** Fluence dependence of ΔmOD at 580 nm (± 20 nm) and 700 nm (± 20 nm) measured 100 ps (± 20 ps) after photoexcitation at room temperature for CsCoW and CoW. The errors of the fluence and ΔmOD values were estimated from the experimental stability of pump light ±5% and the calculated standard deviations of averaged values in the areas in the range of ±20 ps and 580 nm ±20 nm (or 700 nm ±20 nm). **f** Schematic representation of the photo-induced dynamics in cyanido-bridged cobalt-tungstate assemblies. Photoexcitation of the LT state (blue) induces the local charge-transfer-induced spin transition (CTIST) within 130 fs to the PI state (red). At high fluence and elevated temperatures, thermoelastic conversion occurs in CsCoW.

formation near the thermal phase transition temperatures (at RT and 230 K) under high laser fluence. The maximum of this transient absorption band gradually shifts toward shorter wavelengths and decays within 800 ps (Fig. 4a, Supplementary Fig. 14a, b). This shift can be attributed to lattice relaxation affecting the Co–N bonds, influencing the MMCT between $Co^{II}_{HS}$ and $W^V$ owing to the larger $e_g$-$t_{2g}$ energy gap in the Co ligand field compared to the original HT state. The dynamics exhibit strong temperature dependence. At low temperature (100 K) in CsCoW, $\Delta OD(t)$ is dominated by the pronounced decrease of the MMCT band of the LT state, which masks the PI state band (Supplementary Figs. 15, 16, Supplementary Tables 4 and 5). The OD change map at 100 K closely resembles CoW at RT (Fig. 4a), where no thermal phase transition occurs. Notably, slower conversion is observed at 230 K and RT only under high fluence. These features of the slower conversion are characteristic of the so-called thermoelastic conversion, as recently reported for spin-crossover materials[58,59].

## Thermoelastic conversion

We next investigated the fluence dependence of the photoresponse in CsCoW and CoW at RT to examine the thermoelastic conversion (Fig. 4d, e, Supplementary Figs. 17, 18). In CsCoW, $\Delta OD(t)$ around 600 nm is positive and increases with higher pump fluence. In contrast, for CoW, $\Delta OD(t)$ around 600 nm is negative and decreases consistently with fluence. The OD changes at 580 and 700 nm, measured 100 ps after photoexcitation, are shown in Fig. 4d, e. The nonlinear increase in $\Delta OD(t)$ for CsCoW indicates a cooperative process driving the PI state, while the linear decrease in $\Delta OD(t)$ for CoW reflects a local process dominated by the bleaching of the LT MMCT band. These observations strongly indicate that the dynamics in CsCoW within 100 ps correspond to thermoelastic conversion, as in the case of spin-crossover materials. Remarkably, when the pump fluence exceeded 13 mJ cm⁻² for CsCoW, no OD change was detected during the time-resolved optical spectroscopy. This result indicates persistent PIPT at RT, transitioning from the LT to PI (HT) branches of the hysteresis.

Overall, the dynamics in the cyanido-bridged cobalt-tungstate assemblies correspond to the thermoelastic conversion from the LT to PI states, driven by two photoinduced phenomena (Fig. 4f). First, local molecular photoswitching occurs within 130 fs, involving the volume change by STs that generate internal pressure, stabilizing the higher-volume PI state. Second, the lattice warming caused by the laser heating facilitates the thermal conversion from the LT to PI states. The thermoelastic conversion exhibits different behavior when the temperature and/or the laser fluence change, as it relates to the thermal population through the energy barriers, which is changing with photoexcitation density[58,59]. Thus, our data align with the physics of thermoelastic processes, where elastic lattice expansion, driven by the local molecular PIPT favors the thermal population of the high-entropy PI state. When the system is far from a thermal phase transition, as is the case for CsCoW at 100 K and CoW at RT, the thermoelastic conversion is weak. However, near the thermal phase transition, as it is for CsCoW at 230 K and RT, the laser pump fluence induces substantial volume strain and temperature jumps, resulting in robust thermoelastic conversion. As highlighted earlier, CsCoW enhances the HT state stability by destabilizing the hydrogen-bond network in the crystal structure through partial $Cs^+$ ion substitution. From a more chemical perspective on cation substitution in the present case, in addition to the loss of hydrogen bonds, it is crucial to control the phase transitions by selecting counterions based on lower acidity and a larger ionic radius. Lower acidity stabilizes $W^V$ over $W^{IV}$[38], while a larger ionic radius promotes a more stable high-volume state due to thermal vibrations between layers. This substitution in CsCoW shifts the thermal phase transition closer to RT by modulating the balance between intermolecular interactions and cooperativity mediated by the cyanido-bridged networks. This enables thermoelastic conversion within the PIPT out-of-equilibrium dynamics in the vicinity of RT.

## Discussion

In conclusion, we investigated the photoinduced CTIST process in cyanido-bridged cobalt-tungstate assemblies by comparing CsCoW and CoW materials. Optical and magnetic measurements at low temperatures for CsCoW confirmed reversible PIPT in CsCoW, accompanied by a dramatic optical color change between blue in the LT state and red in the PI state, resembling the HT state, and photomagnetic behavior. The photomagnetic properties of the PI state in CsCoW exhibited photoinduced magnetization, with a Curie temperature of 27 K and a coercive field of 7500 Oe at 2 K.

We also studied the out-of-equilibrium photoinduced dynamics of CoW and CsCoW using femtosecond optical spectroscopy. The dynamics comprise a molecular sub-ps process and a macroscopic ps thermoelastic conversion. The first step involves, at the molecular level, the PE state resulting from MMCT from $Co^{III}_{LS}$-$W^{IV}$ to $Co^{II}_{LS}$-$W^V$, followed by ST within 130 fs to the PI $Co^{II}_{HS}$-$W^V$ state. This process is characterized by the coherent activation of the breathing mode of the $CoN_6$ core (approximately 130 cm$^{-1}$). These initial CTIST dynamics were observed in both CoW and CsCoW across all temperatures, as the two compounds exhibit similar molecular and electronic structures around the photoactive Co and W sites. This local photoswitching process drives a subsequent thermoelastic step, where molecular expansion within the lattice, coupled with laser heating, balances the relative stabilities of the LT and PI (HT) states. Thermoelastic conversion is consequently observed in the vicinity of the thermal phase transition and at high fluences in CsCoW, consistent with theoretical predictions for other bistable molecular materials[58,59]. Overall, the present work underlines the direct evidence of photoinduced CTIST in cyanido-bridged cobalt-tungstate assemblies directly and highlights the critical role of $Cs^+$ ion substitution, which is not only responsible for shifting the thermal transition towards RT but also allows thermoelastic conversion in PIPT.

## Methods

### Syntheses and characterizations

The crystalline powder of CsCoW was synthesized by mixing 2 mL of an aqueous solution including 0.50 mmol of $Co^{II}Cl_2 \cdot 6H_2O$, 1.00 mmol of 4-bromopyridine hydrochloride, and 2.50 mmol of CsCl with 5 mL of an aqueous solution including 0.50 mmol of $Cs_3[W(CN)_8] \cdot 2H_2O$ and 1.50 mmol of CsCl for 6 h at RT in the dark. The precipitate of CsCoW was obtained by filtering, washing with a small amount of water, and drying overnight in the dark[38]. The crystalline powder of CoW was synthesized by stirring 6 mL of an aqueous solution including 2.00 mmol of $Co^{II}Cl_2 \cdot 6H_2O$ and 6.00 mmol of 4-bromopyridine hydrochloride with 6 mL of an aqueous solution including 2.00 mmol of $Na_3[W(CN)_8] \cdot 2H_2O$ for 1 h at RT in the dark. The precipitate of CoW was obtained by filtering, washing with acetone, and drying overnight in the dark[39]. Characterizations induced CHN analysis, powder X-ray diffraction, IR spectroscopy, UV-vis-NIR spectroscopy, and magnetic measurements (Supplementary Note 1). Elemental analysis results were as follows: CsCoW: Calculated: C, 27.2%; H, 1.6%; N, 16.8%. Found: C, 27.0%; H, 1.8%; N, 16.5%, for CoW: Calculated: C, 26.9%; H, 1.6%; N, 17.4%. Found: C, 26.8%; H, 1.8%; N, 17.3%. Elemental analyses were performed using the standard microanalytical method for C, H, and N. Magnetic properties were studied with a SQUID magnetometer (Quantum Design MPMS). The temperature-dependent magnetic susceptibility measurement was conducted under an externally applied magnetic field of 1000 Oe. Powder X-ray diffraction (PXRD) measurements were performed using a Rigaku Miniflex diffractometer equipped with Cu Kα radiation (λ = 1.5406 Å). The diffraction patterns were recorded in the 5–60° angle range in 0.02° steps with an exposure time of 1° min$^{-1}$. The IR spectra at RT were measured by using a spectrometer (JASCO FT/IR-4100). The powder samples were prepared by dispersing powder in KBr. THz time-domain spectroscopy (TDS) measurements at RT were obtained in the LT state of CsCoW and CoW using Advantest TAS7400TS THz-TDS in the transmittance mode. UV-vis-NIR spectra at RT were performed by using a spectrometer (JASCO V-670). The sample was prepared by dispersing powder in BaSO$_4$.

### Photoinduced phase transition of CsCoW at low temperatures

Photo-induced UV-vis-NIR absorption spectra were measured using a Shimadzu UV-3600 plus spectrometer with an Oxford Instruments Microstate-He cryostat and the cw diode laser of 785 nm and 532 nm. The sample was prepared by dispersing powder in paraffin oil sandwiched between quartz plates. Photomagnetic measurements were conducted using SQUID equipped with an optical fiber of which the edge has connected the sample and cw diode lasers at 785 nm and 532 nm. The sample for photomagnetism was prepared by dispersing the CsCoW powder in the tape.

### Time-resolved optical spectroscopy

Time-resolved pump-probe experiments were conducted at the Institut de Physique de Rennes using a shot-to-shot optical setup (Supplementary Fig. 2). Time-resolved pump-probe setup employed a Ti:sapphire femtosecond laser (Astrella, 800 nm, 7 mJ, 65 fs, 1 kHz). The 800 nm beam was split; the pump was tuned to 850 nm (500 Hz, via chopper), and the probe supercontinuum (450–800 nm)[47] was generated via a sapphire plate. The probe (150 × 150 μm²) and pump (450 × 450 μm²) beams were co-aligned with minimal angular offset. The probe was collected with a spectrometer (Princeton Instruments ACTION SP2500) and detected with a 2D CMOS camera (Basler aca2440-20gm) at 1 kHz. The pump light was set to 850 nm, targeting the tail of the $W^{IV}$-to-$Co^{III}$ MMCT band to maximize the penetration depth. A 750-nm long-pass filter was inserted into the probe path to block the pump beam, allowing only the probe supercontinuum to be monitored. Samples were dispersed in paraffin oil and sandwiched

between thin glass plates. A reference sample was prepared by sandwiching paraffin oil between identical glass plates. Sample temperature was controlled using an Oxford Instruments Cryojet with an $N_2$ cryojet. The measurements at RT were conducted under atmospheric conditions. The cryogenic measurements were performed employing similar sample conditions under $N_2$ gas stream by the Cryojet. In both cases, we did not measure under a vacuum. Data were analyzed using iterative fitting of multiple exponential functions convoluted with a Gaussian instrument response function with a full-width at half maximum of 65 fs (see Supplementary Note 2). For sub-ps analysis, data correction involved chirp compensation (using quadratic fits of time-zero artifacts) and substrate artifact subtraction (scaled 0.8–1.1). Optical responses were decomposed into population and oscillation components, with oscillations isolated by subtracting the fitted population signal after 300 fs. FFT analysis (limited to 500 cm$^{-1}$) revealed dominant modes at ~130 and 58 cm$^{-1}$, with stronger signals around 700 nm. Lower stability at 230 K and 100 K was attributed to cryostat interference and reduced signal strength. Oscillations were fitted using two damped modes (see Supplementary Note 6).

## Calculation

DFT calculations were conducted to elucidate the origin of the oscillations observed in the sub-ps time delay measurements[53]. Since the crystal structures were too complex for phonon mode calculations, we focused on the Co site unit, Co(4-bromopyridine)$_2$(NC)$_4$, in three distinct electronic states: Co$^{III}_{LS}$, Co$^{II}_{LS}$, and Co$^{II}_{HS}$. Geometry optimization and vibrational frequency calculations were performed for all states using the B3LYP 6-31 G$^+$(2df,2p) functional. A restricted spin approach was applied for Co$^{III}_{LS}$, while an unrestricted spin approach was used for Co$^{II}_{LS}$ and Co$^{II}_{HS}$, with computations carried out in Gaussian16. The vibrational frequencies obtained are detailed in Supplementary Note 7.

## Data availability

The supplementary information provides all data generated in this study, including characterizations, experimental configurations, raw data treatment, data fitting for time-resolved optical measurements, and DFT calculations. Source data files are provided with this paper. Relevant data are also available upon request from the authors. Source data are provided with this paper.

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

## Acknowledgements

This work was supported in part by a Grant-in-Aid for Scientific Research (A) from JSPS KAKENHI (Grant Number 20H00369, 25H00866), Advanced Technologies for Carbon-Neutral (ALCA)-Next from JST (JPMJAN23 A2), CNRS-University of Tokyo"Excellence Science" Joint Research Program, Second CNRS–University of Tokyo PhD Joint Program. The Cryogenic Research Center, The University of Tokyo, the Center for NanoLithography & Analysis, The University of Tokyo, and Quantum Leap Flagship Program (Q-LEAP, Grant Number JPMXS0118068681) by MEXT are also acknowledged for support. K. Nakamura was supported by the World-Leading Innovative Graduate Study Program for Materials Research, Information, and Technology (MERIT-WINGS) and the Fellowship for Integrated Materials Science and Career Development. K. Nakabayashi acknowledges the Iketani Science and Technology Foundation (Grant Number 0351111-A). The authors gratefully acknowledge the Agence Nationale de la Recherche for financial support under grant ANR-19-CE30-0004 ELECTRO-PHONE, ANR-19-CE29-0018 MULTICROSS. E.C. thanks the University of Rennes, the Fondation Rennes 1, and Region Bretagne (Boost'ERC) for funding.

## Author contributions

K.Nakamura, L.G., K.Nakabayashi, E.C., and S.O. conceived the project. K.Nakamura synthesized and characterized samples and performed photoinduced UV-vis-NIR absorption measurements and photomagnetic measurements for **CsCoW** and performed theoretical calculations. K.Nakamura, G.P., and L.G. conducted the transient absorption measurements. K.Nakamura, G.P., L.G., and M.H. analyzed the data. All authors discussed the experimental and theoretical results. K. Nakamura, L.G., K.Nakabayashi, and E.C. wrote the manuscript, with contributions from all authors.

## Competing interests

The authors declare no competing interests.
