## [Transparent Peer Review file · Nature Communications]

Ultrafast charge-transfer-induced spin transition in cobalt-tungstate molecular photomagnets

Corresponding Author: Professor Shin-ichi Ohkoshi

Version 0:

Reviewer comments:

Reviewer #1

(Remarks to the Author)

In this work, the author investigated the ultrafast charge-transfer-induced spin transition (CTIST) process in cobalt-tungstate cyanide-bridged molecular photomagnets. Using femtosecond spectroscopy, the authors revealed the molecular dynamics and thermoelastic conversion mechanisms involved in the photoinduced phase transition from the low-spin state (CoIIIS-WIV) to the high-spin state (CoIIHS-WV). The study revealed that photoexcitation first leads to a transient photoexcited state (CoIIIS-WV), which then transitions to the high-spin state within approximately 130 fs. Additionally, this paper explores the effects of Cs⁺ ion doping on photoinduced phase transitions and verifies the reversibility and stability of these transitions. This is thorough and well-organized work on this interesting molecular material field. The reviewer recommends that the editor consider accepting the article, but the authors should revise and answer the following questions.

1. In the section of introduction, the literature review is not comprehensive, especially with fewer citations of classic studies and recent advances. A comprehensive review of the relevant field should be added in the introduction section, citing more classic and recent research literature to highlight the innovation and significance of this work.
2. In the results and discussion section, while the authors present the CTIST mechanism as the primary explanation, the potential role of the STICT (spin transition-induced charge transfer) mechanism is not fully explored. Although experimental data and theoretical calculations support the CTIST mechanism, a more detailed discussion of the conditions under which the STICT mechanism might arise, or why it is not applicable in this study, would be beneficial. A comparative analysis of these two mechanisms may further strengthen the conclusions and offer a more comprehensive understanding of the system's behavior.
3. In the study of the mechanism, the author focused primarily on the specific conditions of temperature and laser energy density and seldom addressed behaviors under other conditions. Is it possible that the CTIST mechanism remains applicable at higher or lower temperatures or at different laser wavelengths? Whether the author can adequately discuss the influence of these conditions on the phase transition mechanism may significantly enhance the universality of the research findings.
4. In this work, the authors used femtosecond spectroscopy to observe the rapid decay and spin transition of the excited state of light; however, the mechanism of energy transfer and dissipation in these processes is not fully understood. For example, how is the energy of the excited state of light transferred inside the molecule and in the lattice? Are these processes affected by the material structure?
5. Figures 2 and 4 (such as the femtosecond spectroscopy time-resolved maps) present data in a complex manner that may be difficult for nonspecialist readers to understand. Additionally, key data (such as photoinduced phase transition kinetics parameters at different temperatures) are not clearly labeled. The author should optimize the design of figures by adding necessary annotations and explanations to make the data more comprehensible.

Reviewer #2

(Remarks to the Author)

At present the number of reported iron-cobalt compounds possessing different nuclearity and demonstrating charge transfer induced spin transitions (CTIST) accompanied by the conversion of diamagnetic FeII-Is-CN-CoIII-Is units into paramagnetic FeIII-Is-CN-CoII-hs ones via electron transfer is sufficiently high. At the same time the crucial condition for practical applications of systems exhibiting such type transitions is the presence of magnetic bistability that is mainly driven by the cooperativity induced by intermolecular interactions in the crystal lattice. Namely, the useful property of bistability governs

and pushes forward the ongoing search of new solid state magnetic materials containing other than cobalt and iron ions which could demonstrate charge transfer induced spin transitions. Moreover, the study of solid-state phase transitions accompanied by electron transfer between the constituent parts of the material is one of the subjects which lies at the very heart of modern materials science. From this point of view the topic of the paper under consideration that examines for the first time CTIST in two new cyanido-bridged compounds CsCoW and CoW manifesting photo-induced charge transfer and containing a combination of transition metal ions other than iron-cobalt is actual. The authors demonstrate that under action of light it is possible to convert the Co-W pair from the $ls\text{-}CoIII\text{-}WIV$ state to the $hs\text{-}CoII\text{-}WV$ state. With the aid of femtosecond optical spectroscopy and magnetic measurements they were in success to reproduce the course of the spin transformation and to reveal the steps through which the system passes to reach the $hs\text{-}CoII\text{-}WV$ state and, namely, the first step, in which the system undergoes the transformation from $ls\text{-}CoIII\text{-}WIV$ to $ls\text{-}CoII\text{-}WV$ and the second one including the transformation $ls\text{-}CoII\text{-}WV$ \rightarrow $hs\text{-}CoII\text{-}WV$ accompanied by the spin transition in the $CoII$ -ion. In the paper this transformation is also confirmed by DFT calculations. From my point of view the paper deserves publication because it reports charge transfer induced spin transitions in new systems and presents with the aid of femtosecond optical spectroscopy solid experimental proofs for occurrence of this type transitions. The experimental findings are supplemented by DFT calculations, which support the interpretation of the observed phenomena. Undoubtedly, the paper contains new scientific results which, I hope, will be of interest for the wide audience working in the field of photoinduced magnetism as well as for further practical applications. However, in spite of this positive appreciation of the results obtained by the authors several remarks should be made which do not concern the main idea of the paper.

1. Abstract, line 19, it is desirable to correct the phrase "In photoinduced phase transition materials, where both charge transfer and spin transition occur, there has been a long debate on which of the two processes are leading the phase transition"
2. Abstract, lines 23-24, the sentence "Optical and magnetic investigations reveal that the optical excitation ..." should be corrected.
3. p.3, lines 53-56, the sentence should be corrected, it will not be clear for the reader.
4. The meaning of all abbreviations should be explained in the place they first appear.
5. It is desirable to have a more transparent explanation of the slight red shift of the metal-metal charge transfer band in the photo-induced state compared to the band at 495 nm in the high-temperature state (p.5)
6. p.6, lines 108-110, the following statement made in the paper needs more clarification: "the magnetization increases due to the ferromagnetic nature of the PI state, characterized by a Curie temperature (TC) of 27K. Since TC is identical to that reported for CoW, we conclude that the superexchange interaction between Co and W in CsCoW and CoW has similar strength, as both share a similar cyanide-bridged layer structure"
7. In the paper the fitting of the optical density change $\Delta OD(t)$ was performed using an exponential decay model for the transient PE state, and on this basis a conclusion was made that this state decays within approximately 130 fs into the PI state. How is valid the employed approximation for the description of the decay of the PE state?
8. It will be nice if the phrase "Unique breathing modes were identified around 130 cm^{-1} in $CoIIIS$ and $CoIIHS$, based on the optimization of $CoIIIS$ " (p.9, lines 186-188) is explained in more detail. In fact in systems with labile electronic states the breathing modes in the ls - and hs - states differ. One also should bear in mind that the B3LYP functional used for calculations of the vibrational modes in the present paper is not so good for their estimation. Moreover, I doubt if in a paper submitted for publication in "Nature Communications" it is necessary to list in its main text the description of calculations of the vibrational modes by the DFT method.
9. The caption for Fig. 3a in the text of the paper "Charge density maps of the valence (left) and conduction (right) bands in the LT state, reproduced from reference 36 with permission from RSC. Optical excitation ($h\nu$) of the ground LT $CoIIIS\text{-}WIV$ state leads to the photoexcited (PE) $CoIIIS\text{-}WV$ state by promoting one electron transfer from $WIV(dZ2)$ to $CoIIIS(dZ2)$ orbitals" does not correspond to the content of this figure.

Reviewer #3

(Remarks to the Author)

In the submitted manuscript by Eric and Ohkoshi et al. report the "Ultrafast charge-transfer-induced spin transition in cobalt-tungstate molecular photomagnets". Here, authors have used femtosecond optical spectroscopy to investigate the charge-transfer-induced spin transition (CTIST) process in cyanido-bridged cobalt-tungstate assemblies. In earlier manuscripts detailed optical and magnetic investigations of $(H5O2^+)[Co(4\text{-bromopyridine})_2\{W(CN)_8\}]$ (CoW) and magnetic properties investigation of $Cs_0.1(H5O2^+)_0.9[Co(4\text{-bromopyridine})_2.3\{W(CN)_8\}]$ (CsCoW) were performed. The experimental data demonstrates that photoexcitation of the W-to-Co charge-transfer band generates a transient $CoIIIS\text{-}WV$ state and the photoexcited state relaxes to the $CoIIHS\text{-}WV$ state via a spin transition process similar like Fe-Co PBAs systems. Low-temperature optical and magnetic measurements for CsCoW validated the occurrence of reversible photoinduced phase transitions (PIPT) in this complex. Also, their investigation focused on the non-equilibrium photoinduced dynamics of CoW and CsCoW, employing femtosecond optical spectroscopy. This dynamic behavior includes a molecular sub-picosecond process alongside a macroscopic picosecond thermoelastic conversion. The content of the reported work is interesting and well presented in detail with magnetic and spectroscopic analysis. The manuscript is well organized and the science reported is clear. I do believe that the work will attract the intense interest of the scientific community working in the current field of switchable molecular magnetic materials, and especially in the area of Metal-to-metal electron transfer and photomagnetism. Also, I do believe that the article will receive a respectable number of citations in the future. Thus, in my opinion, this work deserves acceptance and publication in Nature Communication. Although, some minor comments/suggestions are to be taken into account by the authors before publication.

It will be better if the authors provide the reason why Cs⁺ ion was chosen in this case. Is there any specific reason for that or similar effect can be observed if K⁺ or Rb⁺ or any other bulky mono cation was used?

The authors mentioned that "Remarkably, the PI state persisted for at least 1 day after the light was turned off at 3 K." To support this statement, it will be nice if they can provide T vs time plot after photo-irradiation and after switching of the light to show the stability of the photo-induced state.

Also, the authors have mentioned that the assemblies of cyanido-bridged cobalt-tungstate are associated with the thermoelastic transition from the low-temperature (LT) state to the photo-induced (PI) state, which is facilitated by a light-induced process. Since the authors have used the term "thermoelastic" for these assemblies, did the authors investigate the effect of pressure on these assemblies?

Since, these assemblies exhibited reversible photoinduced phase transitions (PIPT), the authors could have performed the variable-temperature absolute reflectivity measurements for CsCoW assembly.

Minor comments:

Authors should add few references in the line number 45 "This research has facilitated advancements in applications, such as photonic actuators, memory devices, and other photonic technologies."

One general question regarding the pump-probe experimental setup. Did the authors perform the measurements under high vacuum? If so, what strategies were implemented to prevent the degradation of the complexes in this environment?"

Version 1:

Reviewer comments:

Reviewer #1

(Remarks to the Author)

In this revision, the authors have effectively addressed the reviewers' comments and made corresponding revisions, significantly improving the quality of the manuscript. The paper now meets the publication standards of Nature Communications. I recommend the editor consider accepting this manuscript.

Reviewer #2

(Remarks to the Author)

From my point of view all questions raised by the referees have obtained detailed and reasonable answers of the authors. As to my personal comments I can confirm that I am satisfied by the author's answers. I think that the paper in the revised version satisfies publication

Reviewer #3

(Remarks to the Author)

In this revised version submitted by Eric and Ohkoshi et al. report the "Ultrafast charge-transfer-induced spin transition in cobalt-tungstate molecular photomagnets", authors have addressed all concerns made by the reviewers (including the synthesis, structural and physical characterization's part). This work is now suitable for publication in the Nature Communication journal.

Response to Reviewer 1

We greatly appreciate your high evaluation and constructive comments. We have revised our manuscript according to your suggestions. Below is our response to each of your comments.

Comment 1. In this work, the author investigated the ultrafast charge-transfer-induced spin transition (CTIST) process in cobalt-tungstate cyanido-bridged molecular photomagnets. Using femtosecond spectroscopy, the authors revealed the molecular dynamics and thermoelastic conversion mechanisms involved in the photoinduced phase transition from the low-spin state ($\text{Co}^{\text{III}}_{\text{LS}}\text{-W}^{\text{IV}}$) to the high-spin state ($\text{Co}^{\text{II}}_{\text{HS}}\text{-W}^{\text{V}}$). The study revealed that photoexcitation first leads to a transient photoexcited state ($\text{Co}^{\text{II}}_{\text{LS}}\text{-W}^{\text{V}}$), which then transitions to the high-spin state within approximately 130 fs. Additionally, this paper explores the effects of Cs^+ ion doping on photoinduced phase transitions and verifies the reversibility and stability of these transitions. This is thorough and well-organized work on this interesting molecular material field. The reviewer recommends that the editor consider accepting the article, but the authors should revise and answer the following questions.

Answer 1. We deeply appreciate your positive evaluation of our work.

Comment 2. In the section of introduction, the literature review is not comprehensive, especially with fewer citations of classic studies and recent advances. A comprehensive review of the relevant field should be added in the introduction section, citing more classic and recent research literature to highlight the innovation and significance of this work.

Answer 2. We thank the reviewer for this suggestion. We have modified the introduction accordingly and introduced citations of classic studies on CT materials and more recent advances in the study of the ultrafast phenomena. We modified the introduction to the following sentence and cited the classic and recent references.

Page 3 line 39 from the bottom:

Photoinduced phase transitions (PIPTs) represent a promising avenue for switching physical properties by altering electronic and structural degrees of freedom under photoirradiation.¹⁶⁻¹⁸ Ultrafast time-resolved optical techniques allow to gain substantial knowledge in the understanding of the photoinduced processes at play in PIPTs, enabling the study of electronic and structural dynamics¹⁹⁻²² at the molecular scale and the complex and multiscale out-of-equilibrium transformations at the macroscopic scale.²³⁻²⁵

Page 4 line 66 from the top:

In PIPT materials involving both CT and/or ST,⁴⁰⁻⁴² there has been a longstanding debate spanning approximately 30 years on which of the two processes is leading the phase transition: Charge-transfer-induced Spin transition (CTIST), or vice versa (Spin transition-induced Charge transfer: STICT)? Ultrafast pump-probe techniques were used to investigate the photoinduced process.⁴³⁻⁴⁵

Classic papers:

(17) Gütlich, P., Hauser, A. & Spiering, H. Thermal and Optical Switching of Iron(II) Complexes. *Angew. Chem. Int. Ed. Engl.* **33**, 2024–2054 (1994).

(40) Bleuzen, A. *et al.* Photoinduced Ferrimagnetic Systems in Prussian Blue Analogues $\text{Cl}_x\text{Co}_4[\text{Fe}(\text{CN})_6]_y$ (Cl = Alkali Cation). 1. Conditions to Observe the Phenomenon. *J. Am. Chem. Soc.* **122**, 6648–6652 (2000).

(41) Verdaguer, M. *et al.* Molecules to build solids: high T_c molecule-based magnets by design and

recent revival of cyano complexes chemistry. *Coord. Chem. Rev.* **190–192**, 1023–1047 (1999).

(42) Aguilà, D., Prado, Y., Koumoussi, E. S., Mathonière, C. & Clérac, R. Switchable Fe/Co Prussian blue networks and molecular analogues. *Chem. Soc. Rev.* **45**, 203–224 (2016).

Recent papers:

(43) Barlow, K. & Johansson, J. O. Ultrafast photoinduced dynamics in Prussian blue analogues. *Phys. Chem. Chem. Phys.* **23**, 8118–8131 (2021).

(44) Reinhard, M. *et al.* Observation of a Picosecond Light-Induced Spin Transition in Polymeric Nanorods. *ACS Nano* **18**, 15468–15476 (2024).

(45) Vinci, D. *et al.* Capturing ultrafast molecular motions and lattice dynamics in spin crossover film using femtosecond diffraction methods. *Nat. Commun.* **16**, 2043 (2025).

Comment 3. In the results and discussion section, while the authors present the CTIST mechanism as the primary explanation, the potential role of the STICT (spin transition-induced charge transfer) mechanism is not fully explored. Although experimental data and theoretical calculations support the CTIST mechanism, a more detailed discussion of the conditions under which the STICT mechanism might arise, or why it is not applicable in this study, would be beneficial. A comparative analysis of these two mechanisms may further strengthen the conclusions and offer a more comprehensive understanding of the system's behavior.

Answer 3. We thank the reviewer again for the constructive comment. Exploring the ultrafast STICT mechanism requires the possibility to drive by light excitation the spin transition on a metal ion [ref: Cammarata, M. *et al.* Charge transfer driven by ultrafast spin transition in a CoFe Prussian blue analogue. *Nat. Chem.* **13**, 10–14 (2021).]. Another report investigated the crossover from CTIST to STICT processes in the MnFe Prussian Blue Analogue by changing the photoexcited wavelengths [ref: Azzolina, G. *et al.* Exploring Ultrafast Photoswitching Pathways in RbMnFe Prussian Blue Analogue. *Angew. Chem. Int. Ed.* **60**, 23267–23273 (2021).]. In the present work, we could not explore the possibility of STICT in our Co-W molecular assemblies because they do not have the required Co-centered d-d transitions in the range of the vis-NIR region, which we used for photo-excitation. Figure A below shows the results of band energy calculations obtained by DFT. The highest occupied crystalline orbital (HOCO) has a strong W character, while the lowest unoccupied crystalline orbital (LUCO) has a strong Co character. Excitation in the vis-NIR region corresponds to the W → Co optical transition. Co → Co centered transitions are found around 5 eV (250 nm), which we can't reach with our pump beam setup.

Figure A. Band structure (left) and density of states (DOS) (right) of CoW from -7 eV to 5 eV reproduced from reference 38. Grey, blue, and red areas represent total DOS and partial DOS for Co^{III} 3d and W^{IV} 5d, respectively. The W → Co transition is around 1.5 eV (Pink arrow ≈ 800 nm), while

the Co \rightarrow Co transition is around 5 eV (Purple arrow \approx 250 nm).

This discussion is included in Supplementary Note 7 as the following sentence with Figure A as Supplementary Fig. 13.

Supplementary Note 7, page S18, from the top

Supplementary Fig. 13 shows the results of band energy calculations of the LT state, obtained by DFT. The highest occupied crystalline orbital (HOCO) has a strong W character, while the lowest unoccupied crystalline orbital (LUCO) has a strong Co character. Excitation in the NIR region corresponds to the W \rightarrow Co optical transition, which is driving the **CTIST** process. Investigating the STICT process requires using a Co \rightarrow Co centered transitions, which is found around 5 eV (250 nm). We could not explore the STICT mechanism, as our pump beam could not reach this excitation wavelength.

We have changed and added the following sentence in the result and discussion.

Page 5 line 84 from the bottom:

The absorption band is comparable to the MMCT band of CoW, where the highest occupied crystalline orbital (HOCO) consists of d_z^2 orbitals of W^{IV} and p orbitals of nitrogen (N) atoms. In contrast, the lowest unoccupied crystalline orbital (LUCO) comprises d_z^2 orbitals of Co^{III} and sp orbitals of N atoms, as indicated by DFT calculations.³⁸

Page 10 line 197 from the top:

Based on femtosecond optical spectroscopy and DFT calculations, we propose that the photoinduced MMCT process involves a ground LT $Co^{III}_{LS}-W^{IV}$ and a PE $Co^{II}_{LS}-W^V$ states, because the STICT mechanism should be photoexcited the d-d transition of each metals, however, there are no such absorption bands in the vis-NIR region, confirmed by the DFT calculations (Fig. 3a, Supplementary Fig. 13).³⁸

Comment 4. In the study of the mechanism, the author focused primarily on the specific conditions of temperature and laser energy density and seldom addressed behaviors under other conditions. Is it possible that the CTIST mechanism remains applicable at higher or lower temperatures or at different laser wavelengths? Whether the author can adequately discuss the influence of these conditions on the phase transition mechanism may significantly enhance the universality of the research findings.

Answer 4. We thank the reviewer for this comment. As mentioned above, the **CTIST** mechanism is driven by the initial optical transition, which has a CT character. This is due to the unique electronic structure of these materials. Due to the optical gap, the conducting band is not thermally populated, and the electronic structure is unchanged by temperature or pressure in the LT $Co^{III}-W^{IV}$ state. Therefore, only a **CTIST** mechanism, associated with the quantum dynamics during the sub-picosecond timescale, can be induced by photoexcitation in the NIR region. On the other hand, the thermoelastic conversion must exhibit different behavior when the temperature and/or the laser fluence change, as it relates to the thermal population through the energy barriers, which is changing with photoexcitation density. This thermoelastic process was already shown in our data and discussed in our paper. We have added the following sentence about this in the result and discussion.

Page 13 line 252 from the top:

The thermoelastic conversion exhibits different behavior when the temperature and/or the laser fluence change, as it relates to the thermal population through the energy barriers, which is changing with photoexcitation density.^{58,59}

Comment 5. In this work, the authors used femtosecond spectroscopy to observe the rapid decay and spin transition of the excited state of light; however, the mechanism of energy transfer and dissipation in these processes is not fully understood. For example, how is the energy of the excited state of light transferred inside the molecule and in the lattice? Are these processes affected by the material structure?

Answer 5. This is a very important question. These photoinduced phenomena in molecular materials have a strong molecular nature since the initial photoexcitation process is localized at the molecular scale. The energy deposition on the molecule, through electronic excitation, drives consecutive electronic and structural dynamics during inter-system crossing, which is mainly driven by electron-phonon coupling. This point was discussed in the theoretical work by van Veenendaal [van Veenendaal, M. Ultrafast intersystem crossings in Fe-Co Prussian blue analogues. *Sci. Rep.* **7**, 6672 (2017).]. Energy is released by electronic reorganization and the coherent and incoherent activation of molecular phonons through electron-phonon or phonon-phonon coupling. Our data reveals that coherent oscillations are activated, while photocrystallographic data underlines the large volume change of the crystal. Therefore, part of the energy is also transferred into work through molecular-lattice phonon coupling, as photoexcitation induces volume expansion. Theoretical calculations show that higher frequency phonons are also involved, but our limited time resolution does not allow us to track the associated dynamics (required time resolution of 30 fs), while optical spectroscopy does not allow to disentangle electronic and structural dynamics, as it was done by ultrafast X-ray techniques in other materials [ref: Cammarata, M. *et al.* Charge transfer driven by ultrafast spin transition in a CoFe Prussian blue analogue. *Nat. Chem.* **13**, 10–14 (2021)., Vinci, D. *et al.* Capturing ultrafast molecular motions and lattice dynamics in spin crossover film using femtosecond diffraction methods. *Nat. Commun.* **16**, 2043 (2025)., and Barlow, K. & Johansson, J. O. Ultrafast photoinduced dynamics in Prussian blue analogues. *Phys. Chem. Chem. Phys.* **23**, 8118–8131 (2021).]. This will be a future project in our fields. We have added the following sentence about this in the result and discussion.

Page 9 line 173 from the bottom:

A theoretical study discussed how the energy deposition on the molecule, through electronic excitation, drives consecutive electronic and structural dynamics on the sub-100 fs timescale and inter-system crossing, and the important role of the electron-phonon coupling.⁴⁸ The energy is released by electronic reorganization and the coherent and incoherent activation of molecular phonons through electron-phonon or phonon-phonon coupling. We could not investigate these faster dynamics in more detail because of our limited time resolution. In addition, optical spectroscopy does not allow the disentanglement of electronic and structural dynamics, as reported by ultrafast X-ray techniques in other materials.⁴⁴⁻⁴⁶

We have cited the following reference.

(48) van Veenendaal, M. Ultrafast intersystem crossings in Fe-Co Prussian blue analogues. *Sci. Rep.* **7**, 6672 (2017).

On the other hand, we have removed the following original reference 51.

Enachescu, C. *et al.* Theoretical approach for elastically driven cooperative switching of spin-crossover compounds impacted by an ultrashort laser pulse. *Phys. Rev. B* **95**, 224107 (2017).

Comment 6. Figures 2 and 4 (such as the femtosecond spectroscopy time-resolved maps) present data in a complex manner that may be difficult for nonspecialist readers to understand. Additionally, key data (such as photoinduced phase transition kinetics parameters at different temperatures) are not clearly labeled. The author should optimize the design of figures by adding necessary annotations and explanations to make the data more comprehensible.

Answer 6. Thank you very much for the comment. Along with the reviewer's comment, we have added the modification in Figures 2ab, and 4a and the following sentences in the caption of Figures 2ab and 4a, respectively.

Page 25 line 500 from the bottom:

a, b Time delay ΔmOD ($\Delta OD \times 10^{-3}$) maps measured at room temperature for CsCoW and CoW. Each map consists of three parameters: wavelength on the horizontal axis, delay time on the vertical axis, and ΔmOD as a color bar. The negative and positive delays indicate before and after the photoexcitation. The red and blue dashed lines represent the time traces at 600 nm and 700 nm, respectively.

Page 26 line 518 from the bottom:

a Time delay ΔmOD ($\Delta OD \times 10^{-3}$) maps for CsCoW at room temperature, 230, and 100 K and CoW at room temperature (8 mJ cm^{-2}). Each map consists of three parameters: wavelength on the horizontal axis, delay time on the vertical axis, and ΔmOD as a color bar. The negative and positive delays indicate before and after the photoexcitation. The red and blue dashed lines represent the time traces at 580 nm and 700 nm, respectively.

Response to Reviewer 2

We greatly appreciate your high evaluation and constructive comments. We have revised our manuscript according to your suggestions. Below is our response to each of your comments.

Comment 1. At present the number of reported iron-cobalt compounds possessing different nuclearity and demonstrating charge transfer induced spin transitions (CTIST) accompanied by the conversion of diamagnetic $\text{Fe}^{\text{II}}_{\text{ls}}\text{-CN-Co}^{\text{III}}_{\text{ls}}$ units into paramagnetic $\text{Fe}^{\text{III}}_{\text{ls}}\text{-CN-Co}^{\text{II}}_{\text{hs}}$ ones via electron transfer is sufficiently high. At the same time the crucial condition for practical applications of systems exhibiting such type transitions is the presence of magnetic bistability that is mainly driven by the cooperativity induced by intermolecular interactions in the crystal lattice. Namely, the useful property of bistability governs and pushes forward the ongoing search of new solid state magnetic materials containing other than cobalt and iron ions which could demonstrate charge transfer induced spin transitions. Moreover, the study of solid-state phase transitions accompanied by electron transfer between the constituent parts of the material is one of the subjects which lies at the very heart of modern materials science. From this point of view the topic of the paper under consideration that examines for the first time CTIST in two new cyanido-bridged compounds CsCoW and CoW manifesting photo-induced charge transfer and containing a combination of transition metal ions other than iron-cobalt is actual. The authors demonstrate that under action of light it is possible to convert the Co-W pair from the $\text{ls-Co}^{\text{III}}\text{-W}^{\text{IV}}$ state to the $\text{hs-Co}^{\text{II}}\text{-W}^{\text{V}}$ state. With the aid of femtosecond optical spectroscopy and magnetic measurements they were in success to reproduce the course of the spin transformation and to reveal the steps through which the system passes to reach the $\text{hs-Co}^{\text{II}}\text{-W}^{\text{V}}$ state and, namely, the first step, in which the system undergoes the transformation from $\text{ls-Co}^{\text{III}}\text{-W}^{\text{IV}}$ to $\text{ls-Co}^{\text{II}}\text{-W}^{\text{V}}$ and the second one including the transformation $\text{ls-Co}^{\text{II}}\text{-W}^{\text{V}} \rightarrow \text{hs-Co}^{\text{II}}\text{-W}^{\text{V}}$ accompanied by the spin transition in the Co^{II} ion. In the paper this transformation is also confirmed by DFT calculations. From my point of view the paper deserves publication because it reports charge transfer induced spin transitions in new systems and presents with the aid of femtosecond optical spectroscopy solid experimental proofs for occurrence of this type transitions. The experimental findings are supplemented by DFT calculations, which support the interpretation of the observed phenomena. Undoubtedly, the paper contains new scientific results which, I hope, will be of interest for the wide audience working in the field of photoinduced magnetism as well as for further practical applications. However, in spite of this positive appreciation of the results obtained by the authors several remarks should be made which do not concern the main idea of the paper.

Answer 1. We deeply appreciate your positive evaluation of our work.

Comment 2. Abstract, line 19, it is desirable to correct the phrase “In photoinduced phase transition materials, where both charge transfer and spin transition occur, there has been a long debate on which of the two processes are leading the phase transition”

Answer 2. Thank you very much for the comment. Along with the reviewer’s comment, we have changed it to the following sentences in the abstract.

Page 2 line 19 from the top:

In materials exhibiting photoinduced phase transitions, and in which both charge transfer and spin transitions occur, there has long been a debate about which process drives the phase transition.

Comment 3. Abstract, lines 23-24, the sentence “Optical and magnetic investigations reveal that the optical excitation ...” should be corrected.

Answer 3. Thank you very much for the comment. Along with the reviewer’s comment, we have changed it to the following sentences in the abstract.

Page 2 line 23 from the bottom:

Optical and magnetic studies revealed that the photoexcitation of the ground low-temperature (LT) $\text{Co}^{\text{III}}_{\text{LS}}\text{-W}^{\text{IV}}$ state leads to a photoinduced phase transition towards the $\text{Co}^{\text{II}}_{\text{HS}}\text{-W}^{\text{V}}$ state, which is similar to the high temperature (HT) state.

Comment 4. p.3, lines 53-56, the sentence should be corrected, it will not be clear for the reader.

Answer 4. Thank you very much for the comment. Along with the reviewer’s comment, we have changed the following sentences in the introduction.

Page 3 line 53 from the bottom:

These electronic transformations are further associated with the drastic spectral change in the distinct ultraviolet-visible-near-infrared (UV-vis-NIR) absorption due to the metal-to-metal CT (MMCT) from $\text{Co}^{\text{III}}\text{-W}^{\text{IV}}$ to $\text{Co}^{\text{II}}\text{-W}^{\text{V}}$ in the LT or vice versa in the HT states.

Comment 5. The meaning of all abbreviations should be explained in the place they first appear.

Answer 5. Thank you very much for the comment. Along with the reviewer’s comment, we have changed the SQUID to the superconducting quantum interference device in the result and discussion.

Page 5 line 82 from the top:

The PIPT of CsCoW at low temperatures was investigated using UV-vis-NIR absorption spectroscopy and a superconducting quantum interference device magnetometer.

Comment 6. It is desirable to have a more transparent explanation of the slight red shift of the metal-metal charge transfer band in the photo-induced state compared to the band at 495 nm in the high-temperature state (p.5)

Answer 6. Thank you very much for the comment. The slight shift originated from the distortion of the cyanido-bridged layers by volume changing by temperature. The distortion destabilizes the energy of t_{2g} of Co by decreasing the effect from the π -back donation of cyanide ligands. As a result, the MMCT band of the HT state between $d(t_{2g})$ of $\text{Co}^{\text{II}}_{\text{HS}}$ and d_z of W^{V} becomes smaller by the volume contraction at low temperatures. Along with the reviewer’s comment, we have changed the following sentences in the introduction.

Page 5 line 94 from the bottom:

The MMCT band in the PI state is slightly red-shifted compared to the band at 495 nm in the HT state, which is attributed to the smaller energy gap due to changes in the ligand field of the Co site, mainly decreasing the π -back donation from the cyanide ligand, owing to the distortion of the cyanido-bridged network by the volume contraction at low temperatures (Supplementary Fig. 4a).

Comment 7. p.6, lines 108-110, the following statement made in the paper needs more clarification: “the magnetization increases due to the ferromagnetic nature of the PI state, characterized by a Curie temperature (T_C) of 27 K. Since T_C is identical to that reported for CoW, we conclude that the superexchange interaction between Co and W in CsCoW and CoW has similar strength, as both share a similar cyanido-bridged layer structure”

Answer 7. Thank you very much for the comment. The Curie temperature of cyanido-bridged assemblies strongly depends on the superexchange interactions between the spin of the metals through the cyanido-bridged network and their dimensionality, such as one-dimensional chain, two-dimensional layer, and three-dimensional network. In the present case, both show the same Curie temperatures, indicating that both cyanido-bridged networks (layer) are also similar despite the partial substitution of cation between layers. The J value of the superexchange interaction was 8.4 cm^{-1} , as calculated by the molecular field theory. Along with the reviewer’s comment, we have added the following sentences in the result and discussion.

Page 6 line 108 from the top:

Under a 785 nm irradiation in the LT state at 3 K, the magnetization increases due to the ferromagnetic nature of the PI state, characterized by a Curie temperature (T_C) of 27 K, the same as that reported for CoW. Both J values of the superexchange interaction are 8.4 cm^{-1} , calculated by the molecular field theory.³⁶ Since the T_C strongly depends on the superexchange interactions between the metal spins through the cyanido-bridged network and their dimensionality, such as one-dimensional chain, two-dimensional layer, and three-dimensional network, the same T_C indicates the similar superexchange interaction between Co and W in CsCoW and CoW, and remaining similar cyanido-bridged layer structures despite the partial cation substitution.

Comment 8. In the paper, the fitting of the optical density change $\Delta OD(t)$ was performed using an exponential decay model for the transient PE state, and on this basis a conclusion was made that this state decays within approximately 130 fs into the PI state. How is valid the employed approximation for the description of the decay of the PE state?

Answer 8. Thank you very much for the comment. This question is directly related to comment 5 of reviewer 1. As explained above, intersystem crossing involves electronic and structural dynamics, which are mainly driven by electron-phonon coupling, and results in electronic reorganization and the coherent and incoherent activation of molecular phonons through electron-phonon or phonon-phonon coupling. Our optical spectroscopy data do not allow to disentangle electronic and structural dynamics, and we can only provide a characteristic timescale, considering that the photoexcited state exhibits an exponential decay towards the photoinduced state. This is often used in many ultrafast pump-probe studies. The decay of 130 fs was obtained by averaging all the fitting results of each compound and temperature at 600 nm and 700 nm. In Supplementary Table 1, the error of each component is calculated by the least-squares method, and the fitted values are 118 ± 9 fs of CoW at RT, 136 ± 5 fs of CsCoW at RT, 131 ± 16 fs at 230 K, and 125 ± 19 fs at 100 K, respectively. There is no clear evolution of this decay between CoW and CsCoW or with temperature, and this timescale is comparable with the one found for other spin transition materials. We have added the following sentences in the result and discussion along with the reviewer's comment.

Page 8 line 157 from the bottom:

This decaying timescale is similar for CoW and CsCoW and comparable to that found in ST materials. In addition, it does not depend on the temperature, which is expected, given that it is driven by an electron-phonon coupling process rather than a thermally activated one.⁴⁸

Comment 9. It will be nice if the phrase “Unique breathing modes were identified around 130 cm⁻¹ in Co^{II}_{LS} and Co^{II}_{HS}, based on the optimization of Co^{II}_{LS}” (p.9, lines 186-188) is explained in more detail. In fact in systems with labile electronic states the breathing modes in the ls- and hs- states differ. One also should bear in mind that the B3LYP functional used for calculations of the vibrational modes in the present paper is not so good for their estimation. Moreover, I doubt if in a paper submitted for publication in “Nature Communications” it is necessary to list in its main text the description of calculations of the vibrational modes by the DFT method.

Answer 9. Thank you very much for the important comment. We also understood that the B3LYP function is not the best choice for the phonon mode calculations. Therefore, we confirmed by comparing not only with the experimental results of IR and THz absorption spectra but also calculations with other functions (Lanl2dz) using Gaussian16 based on the LT state, which show the similar phonon modes on the Co site to B3LYP function (Figure B). Along with the reviewer’s comment, we have removed the following sentences from the result and discussion to Supplementary Note 7.

Page 9 line 183 from the bottom:

Meanwhile, modes originating from 4-bromopyridine ligands, including stretching and bending vibrations of the pyridine ring and C-H bonds, were observed in the 600–1600 cm⁻¹ range and around 3200 cm⁻¹, respectively.

Moreover, we have changed to the following sentences in the result and discussion and added the following sentence to Supplementary Note 7.

Page 9 line 182 from the bottom:

Phonon modes arising from cyanide ligands were observed below 500 and around 2100 cm⁻¹, and unique breathing modes were identified around 130 cm⁻¹ in Co^{II}_{LS} and Co^{II}_{HS}, based on the structural optimization of Co^{II}_{LS}. These modes involve the stretching of organic and cyanide ligands. This type of mode is known to be strongly coupled with STs by changing drastic ligand fields between *t*_{2g} and *e*_g orbitals, as the modes act as the reaction coordinate.

Supplementary Note 7, page S15, line 2 from the top

Phonon modes of 4-bromopyridine were observed in the 600–1600 cm⁻¹ and around 3200 cm⁻¹.

Figure B Phonon modes in the THz-region. Black, red, and blue lines indicate the calculation by B3LYP, Lanl2dz, and the experimental data of CsCoW, respectively.

Comment 10. The caption for Fig. 3a in the text of the paper “Charge density maps of the valence (left) and conduction (right) bands in the LT state, reproduced from reference 36 with permission from RSC. Optical excitation ($h\nu$) of the ground LT $\text{Co}^{\text{III}}_{\text{LS}}\text{-W}^{\text{IV}}$ state leads to the photoexcited (PE) $\text{Co}^{\text{II}}_{\text{LS}}\text{-W}^{\text{V}}$ state by promoting one electron transfer from $\text{W}^{\text{IV}}(\text{d}z^2)$ to $\text{Co}^{\text{III}}_{\text{LS}}(\text{d}z^2)$ orbitals” does not correspond to the content of this figure.

Answer 10. Thank you very much for the comment. Along with the reviewer’s comment, we have changed the following sentences in the caption of Figure 3.

Page 26 line 507 from the bottom:

a Charge density maps of the highest occupied crystalline orbital (HOCO, left) and lowest unoccupied crystalline orbital (LUCO, right) in the LT state, reproduced from reference 38 with permission from RSC. The calculated HOCO and LUCO filled the charge density on W and Co orbitals, respectively, indicating that the photoexcitation ($h\nu$) of the ground LT $\text{Co}^{\text{III}}_{\text{LS}}\text{-W}^{\text{IV}}$ state leads to the photoexcited (PE) $\text{Co}^{\text{II}}_{\text{LS}}\text{-W}^{\text{V}}$ state by promoting one electron transfer from W to Co orbitals.

Response to Reviewer 3

We greatly appreciate your high evaluation and constructive comments. We have revised our manuscript according to your suggestions. Below is our response to each of your comments.

Comment 1. In the submitted manuscript by Eric and Ohkoshi et al. report the “Ultrafast charge-transfer-induced spin transition in cobalt-tungstate molecular photomagnets”. Here, authors have used femtosecond optical spectroscopy to investigate the charge-transfer-induced spin transition (CTIST) process in cyanido-bridged cobalt-tungstate assemblies. In earlier manuscripts detailed optical and magnetic investigations of $(\text{H}_5\text{O}_2^+)[\text{Co}(\text{4-bromopyridine})_2\{\text{W}(\text{CN})_8\}]$ (CoW) and magnetic properties investigation of $\text{Cs}^{+0.1}(\text{H}_5\text{O}_2^+)_{0.9}[\text{Co}(\text{4-bromopyridine})_{2.3}\{\text{W}(\text{CN})_8\}]$ (CsCoW) were performed. The experimental data demonstrates that photoexcitation of the W-to-Co charge-transfer band generates a transient $\text{Co}^{\text{II}}_{\text{LS}}\text{-W}^{\text{V}}$ state and the photoexcited state relaxes to the $\text{Co}^{\text{II}}_{\text{HS}}\text{-W}^{\text{V}}$ state via a spin transition process similar like Fe-Co PBAs systems. Low-temperature optical and magnetic measurements for CsCoW validated the occurrence of reversible photoinduced phase transitions (PIPT) in this complex. Also, their investigation focused on the non-equilibrium photoinduced dynamics of CoW and CsCoW, employing femtosecond optical spectroscopy. This dynamic behavior includes a molecular sub-picosecond process alongside a macroscopic picosecond thermoelastic conversion. The content of the reported work is interesting and well presented in detail with magnetic and spectroscopic analysis. The manuscript is well organized and the science reported is clear. I do believe that the work will attract the intense interest of the scientific community working in the current field of switchable molecular magnetic materials, and especially in the area of Metal-to-metal electron transfer and photomagnetism. Also, I do believe that the article will receive a respectable number of citations in the future. Thus, in my opinion, this work deserves acceptance and publication in Nature Communication. Although, some minor comments/suggestions are to be taken into account by the authors before publication.

Answer 1. We deeply appreciate your positive evaluation of our work.

Comment 2. It will be better if the authors provide the reason why Cs^+ ion was chosen in this case. Is there any specific reason for that or similar effect can be observed if K^+ or Rb^+ or any other bulky mono cation was used?

Answer 2. Thank you very much for the comment. As in previous research, we focused on the relationship between thermal phase transition and intermolecular interactions by comparing $\text{Cs}[\text{Co}(\text{3-cyanopyridine})_2\{\text{W}(\text{CN})_8\}]\cdot 2\text{H}_2\text{O}$, which undergoes a thermal phase transition around 200 K, with $(\text{H}_5\text{O}_2^+)[\text{Co}(\text{4-bromopyridine})_2\{\text{W}(\text{CN})_8\}]$, which remains in the low-temperature (LT) state above room temperature (RT). They have similar two-dimensional layered structures with cations between the layers. To control the thermal phase transition, we attempted to substitute the (H_5O_2^+) cation with Cs^+ in $(\text{H}_5\text{O}_2^+)[\text{Co}(\text{4-bromopyridine})_2\{\text{W}(\text{CN})_8\}]$. Additionally, we synthesized an analogue using Rb^+ , which exhibited a thermal phase transition. However, these Rb-based compounds showed more gradual phase transitions and contained impurities, as detected by powder X-ray diffraction measurements. Furthermore, we attempted to synthesize an analogue with NH_4^+ , but it was not obtained due to its lower incorporation priority compared to the H_5O_2^+ ion. Overall, we concluded that selecting an appropriate cation is crucial, considering both lower acidity and a larger ionic radius. Otherwise, the substitution has minimal impact on phase transitions or results in the formation of different crystal structures. Along with the reviewer’s comment, we have changed the following

sentences in the result and discussion.

Page 12 line 259 from the bottom:

From a more chemical perspective on cation substitution in the present case, in addition to the loss of hydrogen bonds, it is crucial to control the phase transitions by selecting counterions based on lower acidity and a larger ionic radius. Lower acidity stabilizes W^V over W^{IV} ,³⁸ while a larger ionic radius promotes a more stable high-volume state due to thermal vibrations between layers. This substitution in CsCoW shifts the thermal phase transition closer to RT by modulating the balance between intermolecular interactions and cooperativity mediated by the cyanido-bridged networks. This enables thermoelastic conversion within the PIPT out-of-equilibrium dynamics in the vicinity of RT.

Comment 3. The authors mentioned that “Remarkably, the PI state persisted for at least 1 day after the light was turned off at 3 K.” To support this statement, it will be nice if they can provide χT vs time plot after photo-irradiation and after switching off the light to show the stability of the photo-induced state.

Answer 3. Thank you very much for the comments. The sentence indicates that the M - H measurement was not affected by time relaxations because it took almost a day. We attached the magnetization vs time plot under 100 Oe and FCM curves after a day after irradiation, indicating keeping the HT state after a day (Figure C).

Figure C. **a** Magnetization vs T plots of the PI state of CsCoW under 100 Oe. Black, opened-red, and closed-red circles indicate before, during, and after photo irradiation (785 nm, 120 mW cm⁻², 30 min.). **b** FCM curves under 100 Oe before and after photoirradiation and after thermal annealing. Black, red, and opened-black circles indicate before photo irradiation, a day after irradiation, and after thermal annealing at 100 K for 10 min.

Comment 4. Also, the authors have mentioned that the assemblies of cyanido-bridged cobalt-tungstate are associated with the thermoelastic transition from the low-temperature (LT) state to the photo-induced (PI) state, which is facilitated by a light-induced process. Since the authors have used the term “thermoelastic” for these assemblies, did the authors investigate the effect of pressure on these assemblies?

Answer 4. Thank you very much for the comment. We investigated the effect of pressure on CsCoW by powder X-ray diffraction measurement, which showed the pressure-induced phase transition under around 100 MPa at RT (Figure D). However, the thermoelastic effect originates from intermolecular

pressure. Therefore, we cannot demonstrate the quantitative argument by each result and did not discuss it quantitatively in the paper. It is our next challenge, and it is necessary to measure other various measurements, such as heat capacity and particle size, to discuss the relationship.

Figure D. **a** Powder X-ray diffraction patterns in the range of 5 to 60° measured under atmospheric at RT after adding various pressures at RT. **b** Intensities at 17.3° (HT state, red arrow) and 18.0° (LT state, blue arrow) vs pressure plots. The peaks at 28°, 47°, and 56° are Si peaks of reference.

Comment 5. Since, these assemblies exhibited reversible photoinduced phase transitions (PIPT), the authors could have performed the variable-temperature absolute reflectivity measurements for CsCoW assembly.

Answer 5. Thank you for the comment. In the present work, we unified the method to the previous research of CsCoW, which was measured by transmittance measurements by diluting the powder sample to liquid paraffin. It is a good suggestion as an additional method.

Comment 6. Authors should add few references in the line number 45 “This research has facilitated advancements in applications, such as photonic actuators, memory devices, and other photonic technologies.”

Answer 6. Thank you very much for the comment. Along with the reviewer’s comment, we have put the reference of 17 and added the following reference in the following sentence in the introduction.

Page 3 line 44 from the bottom:

This research has facilitated advancements in applications, such as photonic actuators, memory devices, and other photonic technologies.^{17,26}

(26) Maiuri, M., Schirato, A., Cerullo, G. & Della Valle, G. Ultrafast All-Optical Metasurfaces: Challenges and New Frontiers. *ACS Photon.* **11**, 2888–2905 (2024).

Comment 7. One general question regarding the pump-probe experimental setup. Did the authors perform the measurements under high vacuum? If so, what strategies were implemented to prevent the degradation of the complexes in this environment?"

Answer 7. Thank you very much for the comment. The measurements of pump-probe spectroscopy

at RT were conducted under atmospheric conditions. The cryogenic measurements were conducted employing the similar sample conditions under gas stream by the Cryojet using liquid N₂. In both cases, we did not measure them under a vacuum. This is indicated in the method section by the following sentence.

Page 15 line 306 from the bottom:

The measurements at RT were conducted under atmospheric conditions. The cryogenic measurements were performed employing similar sample conditions under gas stream by the Cryojet using liquid N₂. In both cases, we did not measure under a vacuum.

Response to Reviewer 1

We greatly appreciate your evaluation and comments. Below is our response to your comments.

Comment 1. In this revision, the authors have effectively addressed the reviewers' comments and made corresponding revisions, significantly improving the quality of the manuscript. The paper now meets the publication standards of Nature Communications. I recommend the editor consider accepting this manuscript.

Answer 1. We deeply appreciate your comments and evaluation of our work.

Response to Reviewer 2

We greatly appreciate your evaluation and comments. Below is our response to your comments.

Comment 1. From my point of view all questions raised by the referees have obtained detailed and reasonable answers of the authors. As to my personal comments I can confirm that I am satisfied by the author's answers. I think that the paper in the revised version satisfies publication

Answer 1. We deeply appreciate your comments and evaluation of our work.

Response to Reviewer 3

We greatly appreciate your evaluation and comments. Below is our response to your comments.

Comment 1. In this revised version submitted by Eric and Ohkoshi et al. report the "Ultrafast charge-transfer-induced spin transition in cobalt-tungstate molecular photomagnets", authors have addressed all concerns made by the reviewers (including the synthesis, structural and physical characterization's part). This work is now suitable for publication in the Nature Communication journal.

Answer 1. We deeply appreciate your comments and evaluation of our work.